# Challenges in the real world use of classification accuracy metrics: From recall and precision to the Matthews correlation coefficient

**Giles M. Foody** ⓘ *

School of Geography, University of Nottingham, Nottingham, Nottinghamshire, United Kingdom

* giles.foody@nottingham.ac.uk

**Data Availability Statement:** All relevant data are within the paper and its Supporting Information files.

**Funding:** The authors received no specific funding for this work.

## Abstract

The accuracy of a classification is fundamental to its interpretation, use and ultimately decision making. Unfortunately, the apparent accuracy assessed can differ greatly from the true accuracy. Mis-estimation of classification accuracy metrics and associated mis-interpretations are often due to variations in prevalence and the use of an imperfect reference standard. The fundamental issues underlying the problems associated with variations in prevalence and reference standard quality are revisited here for binary classifications with particular attention focused on the use of the Matthews correlation coefficient (MCC). A key attribute claimed of the MCC is that a high value can only be attained when the classification performed well on both classes in a binary classification. However, it is shown here that the apparent magnitude of a set of popular accuracy metrics used in fields such as computer science medicine and environmental science (Recall, Precision, Specificity, Negative Predictive Value, J, $F_1$, likelihood ratios and MCC) and one key attribute (prevalence) were all influenced greatly by variations in prevalence and use of an imperfect reference standard. Simulations using realistic values for data quality in applications such as remote sensing showed each metric varied over the range of possible prevalence and at differing levels of reference standard quality. The direction and magnitude of accuracy metric mis-estimation were a function of prevalence and the size and nature of the imperfections in the reference standard. It was evident that the apparent MCC could be substantially under- or over-estimated. Additionally, a high apparent MCC arose from an unquestionably poor classification. As with some other metrics of accuracy, the utility of the MCC may be overstated and apparent values need to be interpreted with caution. Apparent accuracy and prevalence values can be mis-leading and calls for the issues to be recognised and addressed should be heeded.

**Competing interests:** The authors have declared that no competing interests exist.

## 1. Introduction

The value and use of a classification is dependent on its accuracy. Many metrics of accuracy are available to express different aspects of classification quality and have well-defined properties. Unfortunately, with many metrics some key properties such as independence of prevalence and the meaning of calculated values are defined on the assumption that a gold standard reference is used in the accuracy assessment. This assumption is often untenable in real world applications. The use of an imperfect reference standard and can lead to substantial mis-estimation of classification accuracy and derived variables such as class prevalence.

This article adds to the literature on accuracy assessment by revisiting fundamental issues with widely used accuracy metrics. The latter range from recall and precision that express the accuracy of the positive class of a binary classification through to the MCC that is claimed to provide a truthful guide to overall classification quality. Using simulated data it is shown that the theoretical independence of prevalence of some popular metrics is not realised and that the apparent accuracy of a classification can differ greatly from the truth when an imperfect reference is used. Estimates of classification accuracy and prevalence are shown to change substantially with variation prevalence and the quality of the reference standard. The direction and magnitude of the biases introduced by the use of an imperfect reference varies as a function of the nature of the errors it contains. Critically, accuracy metrics do not always possess the properties claimed in some of the literature because a fundamental assumption underlying accuracy assessment is unsatisfied. For example, it is shown that a relatively high MCC may be estimated for an essentially worthless classification arising from a classifier with the skill of a coin tosser. Consequently, some claims made about the value and use of MCC, and other metrics, for accuracy assessment are untrue. Researchers need to avoid naïve interpretation of accuracy metrics based directly on estimated apparent accuracy values and act to address challenges in the interpretation and use of a classification. They could, for example, interpret apparent values with care, estimate reference data quality and apply methods to correct for the biases introduced into an analysis through the use of an imperfect reference.

### 1.1 Overview

A critical attribute of a classification is its quality, often described in terms of its accuracy. Classification accuracy reflects the amount of error contained in a classified data set and indicates the classification's fitness for purpose. An unacceptably low accuracy might be taken to be a spur to refine and enhance a classifier (e.g. add additional discriminating variables) until a sufficiently accurate classification is achieved. As such, accuracy can be fundamental to classifier development. Such activity is, for example, central to cross-validation in model development. Additionally, a low accuracy might cause a researcher to discard a classifier as inappropriate and select an alternative that might be superior. Ultimately, the accuracy of a classification provides a guide to the performance of the classifier and impacts on the quality of products derived from the classification. For example, the accuracy of a classification influences estimates of class abundance such as the prevalence of a disease in a typical medical application. Critically, the accuracy of a classification is fundamental to its use and ultimately the inferences and decisions made based upon it. Incorrect evaluations of accuracy, however, can be deceiving and can lead to the questioning of the quality and indeed validity of conclusions drawn from a classification analysis [1, 2].

In principle, accuracy assessment is a straightforward task. The accuracy of a classification is simply an indicator of the amount of error in the labels generated by a classifier such as a diagnostic test. The error can be calculated by comparing the classifier's labels with reality. In practice, the labels predicted by the classifier are compared against those obtained from a

reference standard. Typically, the core focus is on the magnitude of a quantitative accuracy metric; often informed with confidence intervals and rigorous statistical testing [3]. Sometimes a scale may be used to aid interpretation of a metric, with the range of possible values divided up to provide an ordinal scale such as low, medium and high accuracy (e.g. [4]). In many situations, the relative magnitude of accuracy metrics is critical, often relative to a target or threshold value (e.g. [1, 5]) or between a set of results obtained from other studies perhaps based on different samples. None-the-less the key focus in evaluating a classification is typically the magnitude of a calculated metric of accuracy.

There is no single perfect metric of classification accuracy [6, 7]. There are many measures of classification accuracy which typically reflect different aspects of the quality of a classification. With binary classifications, which are the focus of this article, popular approaches include, overall accuracy, recall, specificity, $F_1$ and measures such as the area under the receiver operating characteristics or precision-recall curves [8–12]; the metrics used in this paper are defined in section 1.2. These various measures all convey different information about a classification, each with merits and demerits for a particular application. Consequently, there have been many calls for the use of multiple metrics although this can complicate interpretation [13, 14]. Additionally, a common, albeit unrealistic, desire is to have a single value to summarise a binary classification and recent literature has promoted the Matthews correlation coefficient (MCC) for all researchers and all subjects [11, 15].

The focus of this article is on some of the challenges of interpreting the magnitude of a computed accuracy metric. The interpretation of an accuracy statement is more challenging than it may first seem to be in many real world situations. A fundamental concern is that the magnitude of an accuracy metric is not solely a function of the quality of the classification. The magnitude of an accuracy metric can, for example, also be a function of the population being studied and the specific sample of cases used in its calculation [9] as well as of the quality of the reference standard used [16]. The former issue is associated mostly with the effects of prevalence and the latter with deviation from a true gold standard reference. Both of these variables can result in the apparent accuracy of a classification differing from reality. In this situation, the accuracy assessment is also reduced from an objective and generalizable assessment of classification quality to an assessment relative to only the specific sample of data cases and reference standard used [2, 15]. The apparent accuracy indicated by the magnitude of an accuracy metric may differ greatly from reality. Moreover, the magnitude and direction of the deviation from the true accuracy is a function of the nature of the data set used and of the imperfections that exist in the reference standard [2, 16–20].

Critically, the magnitude of an accuracy metric is not influenced solely by the quality of the classification. This makes comparisons of accuracy metric values, whether to a scale or between classifications, difficult. The challenges introduced by variations in prevalence and the use of an imperfect reference standard are well known but sadly are often not or only poorly addressed [1, 16, 21]. Here, the aim is to revisit some of the fundamental issues for a set of popular accuracy metrics but with particular regard to the MCC that has recently been strongly promoted as an accuracy metric yet actually has some undesirable properties associated with prevalence and use of an imperfect reference standard.

The classes in a classification analysis are often imbalanced in real world applications. With a focus on widely used binary classifications, this situation commonly arises when a class, often the one of particular interest, is rare. The prevalence may also vary, perhaps in space and or time. For example, in studies of Crohn's disease the prevalence varied from 20 to 70% in different sub-groups investigated [22]. Even larger ranges of prevalence can sometimes be expected to occur. For example, in satellite remote sensing of tropical deforestation, deforestation may be relatively rare at the global scale but in small local studies it could be completely

absent while other sites have been completely cleared of forest making the full range of prevalence possible. As a result, imbalanced classes may be common [12, 23]. Moreover, the degree of imbalance can be very large. For example, recent literature highlights this issue with reference made to situations in which the ratio of the number of cases in the majority class to that in the minority class included values such as 2,000:1 [24] and 10,000:1 [12].

Some studies seek to reduce the problem of imbalanced classes by sampling the population to achieve a balanced sample. Alternatively, researchers sometimes artificially balance the sample by use of suitable data augmentation procedures or other means to adjust a sample to achieve a desired level of balance [25–27]. But such approaches are not problem-free. Synthetic minority oversampling data augmentation methods have, for example, the potential to increase biases in the data set and overfit to the minority class [28]. Imbalanced classes are, therefore, common and researchers should really correct estimates of accuracy metrics and derived products such as prevalence for the bias induced by class imbalances [29–32]. Commonly, many seek to address this problem by making a call for the use of accuracy metrics that are believed to be independent of prevalence such as recall, specificity and Youden's J [33–35]. The assumption of independence of prevalence also underpins the use of Bayes theorem in applications such as clinical diagnosis [34, 36]. However, the claimed independence of prevalence linked to these and some other accuracy metrics can disappear if an imperfect reference standard is used in the accuracy assessment.

While more attention has been paid to sample issues such as sampling design and class imbalance than to reference data quality [21], the effects associated with the use of an imperfect reference standard are well known [1]. Despite this situation the negative effects associated with the use of an imperfect reference standard are often ignored or only poorly addressed [21].

The use of a perfect, gold standard, reference data set in which all class labels contained are completely correct is often assumed implicitly in an accuracy assessment. However, such a gold standard may not exist [37, 38] or might be unavailable because it is perhaps too costly or impractical to use [16, 20, 39]. Such situations force the use of an imperfect reference standard. In many studies the reference data arise from expert labelling [22]. Unfortunately, such an approach is far from perfect with the level of disagreement between expert interpreters often large. For example, values of disagreement up to 25% noted have been noted in some medical research [40] and even higher, approximately 30%, in a remote sensing studies [41]. These disagreements arise because there are many error sources [20, 21, 42]. Real world data are often messy. Errors can be anything from a typographical mistake to an issue connected to a random event (e.g. shadow in an image complicating labelling) to systematic errors associated with the skills, training and personal characteristics of the people providing labels and the tools that they use. Errors can range from honest mistakes through poor contributions from spammers to deliberately mis-labeled cases provided with malicious intent [21, 43]. Sometimes imperfections in the reference data are noted and the aim is to use a reference that is expected to be more accurate than the classification under-study [3]. The bottom line is that the reference data set used is often not a true gold standard. However, it is common for the reference standard to be poorly, if at all, discussed and imperfections rarely addressed [1, 16]. Although the deviations from perfection can be large it is important to note that even small errors in the reference data can lead to large bias and mis-estimation from which can follow mis-interpretation and the drawing of incorrect conclusions [1, 16, 44]. Furthermore, the magnitude and direction of mis-estimation varies as a function of the nature of the errors present in the reference standard. The challenges in accuracy assessment should not be ignored and researchers have been urged to take action to address them which includes the generation of error corrected estimates of classification accuracy [20, 29–32] and derived variables such as prevalence [45].

The effects of variations in prevalence and reference data error on accuracy assessment are well known and recent literature has focused on metrics to provide meaningful information. Recent literature has also promoted the use of the MCC in accuracy assessment in all subjects and highlighted merits relative to other popular and widely used measures such as recall, precision and $F_1$ [11, 15]. Key features behind the arguments put forward for the use of the MCC are that it can be extended from binary to multi-class classifications, uses all four elements of a binary confusion matrix and is more informative than other popular accuracy metrics [11, 15]. A key property claimed about the MCC is that a high value can only arise if good results are obtained for all four confusion matrix elements [11] and thus a high score can only be obtained if the classification performed well on both of the classes [46]. Moreover, it is claimed that the MCC is robust to imbalanced data sets [46] although variation with prevalence is known to occur [15]. While the magnitude of the MCC can vary with prevalence workarounds exist and it has been suggested that a metric that is claimed to be unbiased by prevalence, such as J, be used if class imbalance is a concern [15]. Here, it is suggested that in real world applications the magnitude of the MCC, and other metrics such as J, can be difficult to interpret and that the potential of the MCC may be over-stated. This situation arises since classification accuracy is a function of more than just the performance of the classifier. For example, a low apparent accuracy can be obtained from a highly accurate classification. Alternatively, the magnitude of a metric, including the MCC, may be artificially inflated complicating the interpretation of large values. Here, a key aim is to revisit some fundamental issues with accuracy assessment and show that some well-known problems apply to metrics such as the MCC.

## 1.2 Background to accuracy assessment

The quality of a binary classification is typically assessed with the aid of a 2x2 confusion matrix. The latter is simply a cross-tabulation of the labels allocated to a set of cases by a classifier against the corresponding labels in a reference data set. The labels used for the two classes may vary between studies (e.g. change *v* no change, yes *v* no *etc.*) but often take the form of positive *v* negative. It is assumed that the classes are discrete, mutually exclusive and exhaustively defined (i.e. each case lies fully and unambiguously in one of the two classes and no other outcome for a case is possible). The binary confusion matrix comprises just four elements but these, and their marginal values, fully describe the classification (Fig 1).

A range of measures of classification quality may be generated from the confusion matrix to illustrate different aspects of the classification. Commonly, however, there is a general desire for a single value to summarise the accuracy of the classification. Unfortunately, the assessment of classification accuracy can be a more challenging and difficult task than it may first appear. In addition, the challenge is further complicated by the use of different terminology in the vast array of disciplines that require and use accuracy assessment (e.g. [33, 44]). This section aims to cover some of the fundamental issues and introduce the terminology to be used to avoid potential confusion. The discussion is focused on only some of the most popular and widely used classification metrics and does not seek to be exhaustive but to show some key issues and trends. The discussion focuses mainly on popular metrics focused on the positive cases such as recall and precision through to the MCC which has recently been promoted as a standard metric for use in all subjects [15].

The four elements of the confusion matrix show both the correct and incorrect allocations made by the classifier. The cases correctly allocated, and lying in the main diagonal of the matrix, are the true positives (TP, cases that were classed as positive and also have a positive label in the reference data) and true negatives (TN, cases that were classed as negative and also have a negative label in the reference data). The cases incorrectly allocated in the classification

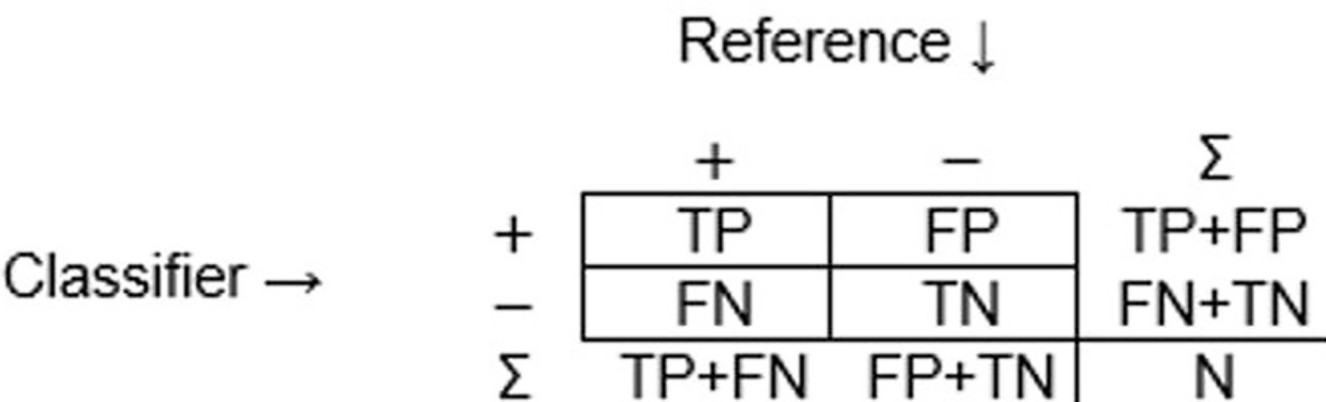

**Fig 1. The binary confusion matrix with positive (+) and negative (-) classes.** Note in the format shown, the labelling provided by the reference data are in the columns and the results of the classifier being assessed are in the rows. Abbreviations used are defined in the text.

are the misclassifications and are false positives (FP, cases labelled positive by the classifier but having a negative label in the reference data) and false negatives (FN, cases labelled negative by the classifier but having a positive label in the reference data). The discussion in this section assumes that the reference data used in determining which element of the confusion matrix to place a classified case into is a true gold standard (i.e. it is perfect, containing zero error).

A simple and widely used metric to express the quality of the entire classification is the proportion of cases correctly allocated which is often referred to as accuracy or overall accuracy which may be calculated from:

$$\text{Accuracy} = \frac{TP + TN}{TP + TN + FN + FP} = \frac{TP + TN}{N} \qquad (1)$$

Although this can be a useful metric that makes use of all four elements of the confusion matrix there are concerns with its use [13]. For example, a key problem is that the metric is strongly impacted by imbalanced classes with a bias toward the majority class and can be uninformative [11, 47]. A variety of other accuracy metrics have been proposed to evaluate the accuracy of binary classifications.

Often an analysis has a focus on the positive cases. An important issue in a classification accuracy assessment is the proportion of positive cases in the reference data set that is often defined as the prevalence (P) and is calculated from:

$$P = \frac{TP + FN}{N} \qquad (2)$$

where N is the total sample size (i.e. the number of cases in the data set, the sum of all the positive and negative cases in the data set). Prevalence is a property of the population under study [9]. The prevalence also indicates the relative balance or size of the two classes in the data set. When prevalence equals 0.5 the classes are balanced with an equal number of positive and negative cases in the reference data set and the ratio of positive:negative cases is 1:1. The more the magnitude of the prevalence deviates from 0.5 greater the degree of imbalance present. In many studies, the positive cases are relatively rare and hence imbalance can become an issue. While some studies seek balance and may achieve this via careful sampling or use of augmentation methods many proceed with imbalanced data sets.

With a focus on the positive cases, a simple metric to quantify the accuracy of the classification is Recall that is calculated from:

$$\text{Recall} = \frac{\text{TP}}{\text{TP} + \text{FN}} \tag{3}$$

Recall indicates how well the classifier labels as positive cases that actually are positive, showing the ability to correctly identify cases that are positive [35, 48]. This measure is widely used in computer science and machine learning but also described by other communities as sensitivity, hit rate, true positive rate and producer's accuracy [3, 9, 48]. Although useful it has limitations such as the provision of no information on the FP cases [47]. Additionally, Recall does not fully capture the full information on the accuracy of the classification with regard to the positive cases. Recall expresses the probability of correctly predicting a positive case [47]. Some researchers may have a different perspective and be interested in the probability that a positive prediction is correct. The suitability of a metric depends on the researcher's needs and the application in-hand. For example, some studies may be more focused on commission rather than omission errors and so researchers may at times wish to focus on the rows rather than the columns of the confusion matrix as determined by their perspective [36, 49–51]. An alternative metric to Recall that is focused on the positive cases is Precision, which is calculated from:

$$\text{Precision} = \frac{\text{TP}}{\text{TP} + \text{FP}} \tag{4}$$

Again, although this is a term widely used in computer science and machine learning other expressions such as the positive predictive value and user's accuracy are used in other disciplines [3].

Should interest focus on the negative cases, two additional metrics may be defined. Similar to Recall for the positive cases, the accuracy of the negative classifications may be expressed by Specificity calculated from:

$$\text{Specificity} = \frac{\text{TN}}{\text{TN} + \text{FP}} \tag{5}$$

Specificity indicates the ability of the classifier to correctly identify negative cases [35, 48]. This metric is sometimes referred to as the true negative rate. Additionally, from the same perspective used in the calculation of Precision for the positive cases, the accuracy of the negative cases can be expressed by the Negative Predictive Values (NPV) that can be obtained from:

$$\text{NPV} = \frac{\text{TN}}{\text{TN} + \text{FN}} \tag{6}$$

The magnitude of the four basic metrics of Recall, Precision, Specificity and NPV are all positively related to the aspect of accuracy that they measure and lie on a 0–1 scale. If a gold standard reference is used in the accuracy assessment, a key feature of Recall and Specificity is that are independent of prevalence while Precision and NPV are dependent on prevalence [35].

Each of these four metrics of accuracy (Eqs 3–6) can be informative and useful. Each does not, however, fully summarise the accuracy of the entire classification; each is based on only two of the four confusion matrix elements [15, 47]. It is, therefore, common for two or more metrics to be used together. For example, Recall and Precision are widely reported together to obtain a fuller characterisation of accuracy than that arising from one alone. Other approaches to more fully characterise a classification have been presented in the literature (e.g. [9, 47]).

An alternative characterisation of classification accuracy can be achieved by combing metrics. For example, a popular approach is to determine Youden's J from:

$$J = \text{Recall} + \text{Specificity} - 1 \tag{7}$$

This metric is sometimes referred to as the true skills statistic and bookmaker informedness [9, 15]. The magnitude of J is related positively to classification accuracy and lies on a scale from -1.0 to 1.0. The metric J is often promoted for use in accuracy assessment since its magnitude is independent of prevalence [15] if a gold standard reference is used in the accuracy assessment.

Another widely used metric that essentially combines the information of two of the basic metrics is $F_1$. Specifically, the $F_1$ metric is the harmonic mean of Recall and Precision and may be calculated from:

$$F_1 = 2 \times \frac{\text{Recall} \times \text{Precision}}{\text{Recall} + \text{Precision}} = \frac{2\text{TP}}{2\text{TP} + \text{FN} + \text{FP}} \tag{8}$$

The magnitude of $F_1$ ranges from 0 if Recall and or Precision are zero to 1.0 if both Recall and Precision indicate a perfect classification. Although widely used as an accuracy metric, $F_1$ is described as being inappropriate for use with imbalanced data sets and its magnitude is dependent on prevalence [11] making its magnitude misleading. In addition, the $F_1$ metric does not use all of the information contained in the confusion matrix [15].

Recently, the MCC has been promoted as a standard metric of classification accuracy for all subjects and data sets. The MCC uses all four elements of the confusion matrix [11, 47] and is calculated from:

$$\text{MCC} = \frac{(\text{TP} \times \text{TN}) - (\text{FP} \times \text{FN})}{\sqrt{(\text{TP} + \text{FP}) \times (\text{TP} + \text{FN}) \times (\text{TN} + \text{FP}) \times (\text{TN} + \text{FN})}} \tag{9}$$

The magnitude of the MCC is positively related to the quality of a classification and lies on a scale from -1.0 to 1.0, with an MCC = 0 indicating an accuracy equivalent to that from a coin-tossing classifier [11]. Although some claims to the MCC being robust or relatively unaffected by class imbalance have been made [52] it is known that the MCC can be impacted by prevalence. However, it has been suggested that workarounds exist for this situation or that a metric such as J should be used if class imbalance is a concern [11, 15].

The MCC has been forcefully promoted as superior to other measures such as accuracy and $F_1$, which are the most widely used measures [11]. For example, Chicco *et al.* (2021) [15] argue that a high accuracy (Eq 1) or $F_1$ (Eq 8) guarantee that two of the basic metrics (Eqs 3–6) are high, a high J guarantees that three of the basic metrics are high while a high MCC guarantees that all four basic metrics are high. Thus, the MCC "produces a high score only if the predictions obtained good results in all four confusion matrix categories" [11, page 1].

Many other metrics of classification accuracy are available. Popular metrics beyond the set defined above include likelihood ratios (LR) and the area under the receiver operating characteristics and/or precision-recall curves. LRs are often calculated with regard to both classes. The positive LR is the ratio of the true positivity rate to the false positivity rate [53] and is calculated from:

$$\text{LR}+ = \frac{\text{Recall}}{(1 - \text{Specificity})} \tag{10}$$

The negative LR is the ratio of the false negative rate to the true negative rate [54] which may be calculated from:

$$\text{LR} - = \frac{(1 - \text{Recall})}{\text{Specificity}} \tag{11}$$

LRs lie on a scale from 0 to infinity. A value of 1 indicates a poor classification in which the probability of a classifier predicting a positive label is the same for cases that belong to the positive class and to the negative class. Classifications that have a high LR+ (>1) and low LR- (<1) demonstrate high discriminating ability [54, 55]. The LRs are typically claimed to be unaffected by prevalence [53].

The receiver operating characteristics and precision-recall curves are also based on the basic metrics of accuracy. The receiver operating characteristics curve is simply a depiction of the relationship between Recall and 1-Specificity. The precision-recall curve is also, as evident from its name, the relationship between Precision and Recall. As these, and other metrics, are essentially based on the basic metrics discussed above they will share some properties, including the degree of sensitivity to variations in prevalence [11, 23, 56]. The core focus in this paper will be the accuracy metrics represented by Eqs 3–9 and prevalence calculated from Eq 2; a limited discussion on LRs will be included to illustrate issues on an additional approach used widely in accuracy assessment.

Throughout this section it has been explicitly assumed that a gold standard reference is used in the accuracy assessment. However, an uncomfortable truth in real world studies is that the reference standard is often imperfect. In such situations, an apparent rather than true confusion matrix is generated and this, together with the associated metrics calculated from it, can differ greatly from the truth.

## 1.3 Use of an imperfect reference

Error in the reference data has a simple effect, in essence it simply moves an affected case from one confusion matrix element to another. This has the effect of altering the magnitude of the entries in the confusion matrix and thereby the magnitude of the accuracy metrics that may be calculated from it.

Unfortunately, even small errors in the reference data can be a source of major mis-estimation of accuracy metrics. Furthermore, the magnitude and direction of the mis-estimation a function of the nature of the errors in the reference data set as well as the prevalence [16–19]. If the classifier's errors are conditionally independent of those in made with the reference standard, the errors are unrelated or independent. In this situation, it is common to find that the magnitude of an accuracy metric is often under-estimated. Commonly, it is impossible to assume independence of errors, especially if the classifier and reference are based on the same phenomenon or process [16, 17, 44].

Different trends may be observed if the errors made by the classifier and the reference standard are conditionally dependent and so tend to occur on the same cases. The direction of mis-estimation can be in either direction depending on the strength of correlation [16]. When the degree of correlation is relatively strong, the error rates in an analysis can be substantially under-estimated and hence accuracy metrics over-estimated [16, 17, 31, 32].

Mis-estimates of a derived property such as prevalence can be in either direction. The magnitude of mis-estimation is dependent on the degree of correlation in the errors and the true prevalence [16].

Critically, the fundamental assumption of the use of a gold standard reference in an accuracy assessment is often unsatisfied. The use of an imperfect reference standard results in the

generation of an apparent confusion matrix which can differ greatly from the true matrix that would be formed with a gold standard reference. Consequently, the metrics estimated from the confusion matrix are also apparent values that may differ from the truth.

## 2. Materials and methods

A simple simulation-based approach was used to explore the effects of variations in prevalence and reference standard error on the magnitude of a suite of accuracy metrics and prevalence from apparent confusion matrices. The focus is on the assessment of the accuracy of classifications of known and constant quality (as defined by Recall, Specificity and J) but with differing prevalence and evaluated using imperfect reference standards of varying quality. For simplicity, simple scenarios in which the classification being assessed and reference standard each had Recall = Specificity were used; the equations given below can be used to explore other scenarios.

With knowledge of a classification's true values of Recall and Specificity together with the prevalence it is possible to generate a confusion matrix. The equations to determine the entries in the four elements of a binary confusion matrix are:

$$TP = P \times Recall \tag{12}$$

$$FP = (1 - P) \times (1 - Specificity) \tag{13}$$

$$FN = P \times (1 - Recall) \tag{14}$$

$$TN = (1 - P) \times Specificity \tag{15}$$

For illustrative purposes, some example matrices will be generated for display. This simply requires multiplying the computed value for each element by the sample size. For this purpose it was assumed that the sample size was N = 1,000. Critically, the above equations allow generation of the actual or true confusion matrix that would be observed if a gold standard reference data set was used. The true value for each of the selected accuracy metrics and prevalence may then be estimated from the confusion matrix using Eqs 2–9.

A wide range of values for Recall and Specificity are reported for reference standards in the literature (e.g. [2, 57, 58]). Here attention was focused initially on a simple scenario in which the outputs of a classification, measured against a gold standard reference, could be summarised as Recall = Specificity = 0.8. As a consequence of these latter values, the classification also had J = 0.6. The values selected are essentially arbitrary but are taken to represent what in many instances would be viewed as a 'good' classification. The values are comparable to others reported in the literature but also allow comparison against a set of imperfect reference data sets that are, as is often desired, more accurate than the classification under evaluation. Again, the magnitudes of the imperfections are arbitrary but here three imperfect reference standards of relatively high, medium and low accuracy were generated. These reference data sets contained 2% (i.e. Recall = Specificity = 0.98), 10% (i.e. Recall = Specificity = 0.90) and 18% error (i.e. Recall = Specificity = 0.82) respectively. Consequently, it was possible to estimate accuracy metrics and prevalence using three imperfect standards of differing quality and know also the true values that would arise from the use of a gold standard reference. To further extend the study, the analyses were undertaken twice, once with independent errors in the reference standard and then again using correlated errors.

Generating the apparent confusion matrices and estimating the magnitude of the apparent accuracy metrics and apparent prevalence was undertaken using approaches used previously

in the literature. Specifically, for the situation in which the errors in the reference standard and classified data are independent the equations presented in [30, 59] were used. In this approach, the apparent confusion matrices were generated using the following equations:

$$TP' = (P \times Recall_R \times Recall_C) + ((1 - P) \times (1 - Specificity_R) \times (1 - Specificity_C)) \tag{16}$$

$$FP' = (P \times Recall_C \times (1 - Recall_R)) + ((1 - P) \times (1 - Specificity_C)) \tag{17}$$

$$FN' = (P \times Recall_R((1 - P) \times (1 - Specificity_R))) \tag{18}$$

$$TN' = P \times (1 - Recall_R) \times (1 - Recall_C) + ((1 - P) \times Specificity_R \times Specificity_C) \tag{19}$$

in which the superscript ' highlights that this is the apparent rather than true value and the subscripts R and C refer to the reference standard and the classification respectively. The apparent values for accuracy metrics and prevalence were then calculated from the apparent confusion matrices using Eqs 2–9.

In the case of correlated errors, the approach discussed by [16] was used. In this approach, the true confusion matrices generated earlier were adjusted to reflect the level of error contained in the reference standard. To do this, the number of positive cases corresponding to the relevant error amount (2%, 10% or 18%) were relabelled to be incorrectly negative in both the classification and the reference data. Similarly, the number of negative cases corresponding to the selected error amount were relabelled to be incorrectly positive in both the classification and the reference data.

The apparent confusion matrices were generated, across the full range of prevalence at 0.05 increments; to avoid complications associated with the extreme values of 0 and 1.0 the actual start and end points of the prevalence scale were 0.01 and 0.99 respectively. In each analysis, the classification being evaluated against any of the reference standards had the same basic properties with Recall = Specificity = 0.8.

Thus, in essence, at each level of prevalence, a confusion matrix was generated using a gold reference standard and accompanied by confusion matrices generated using two imperfect reference standards. A total of six sets of confusion matrices were generated with the imperfect reference data as there were three levels of imperfection (2%, 10% and 18%) for situations in which the errors were independent and then when the errors were correlated. From each apparent confusion matrix a set of standard metrics of accuracy were calculated. These are all apparent values rather than truth as the reference data used to form each apparent confusion matrix are imperfect. The core focus was on the four basic metrics of accuracy (Recall, Precision, Specificity and NPV) and then four important and widely used metrics. The latter were the apparent values of J (suggested as an alternative to MCC if imbalance issues are a concern as claimed to be independent of prevalence), $F_1$, MCC and prevalence. Other metrics can, of course, be estimated and as an example results for LRs will also be presented. Some example confusion matrices will also be provided to aid readers wishing to explore other metrics and issues beyond the scope of this article.

Finally, one further set of simulations was undertaken to help illustrate a specific situation in which the MCC is expected to be over-estimated and complicate the interpretation of a relatively high score. This additional set of simulations was focused on evaluation of an unquestionably poor classification using an imperfect reference standard. The classification to be evaluated had Recall = Specificity = 0.5 and J = 0, values that would be obtained from an unskilled or coin-tossing classifier. The imperfect reference data contained 30% correlated

error (i.e. Recall = Specificity = 0.7). This is a less accurate reference data set than used in the other simulations discussed above but still in the range of values reported in the literature. Thus, in this scenario a dreadful classification is assessed relative to an imperfect but still realistic reference data set.

The supporting information files S1–S3 Tables contain the components of the confusion matrices generated in the scenarios reported. The latter information allows the calculation of all of the accuracy metrics used from the equations provided in section 1.2.

## 3. Results

Confusion matrices generated using a gold standard reference (Recall = Specificity = 1.0) and imperfect reference standards with independent and correlated errors (both with Recall = Specificity = 0.9) for two levels of prevalence are shown in Fig 2.

The relationship between the apparent accuracy indicated by the four basic metrics (Recall, Precision, Specificity and NPV) and prevalence generated from the use of three imperfect reference data sets of differing accuracy associated with inclusion of independent errors is shown in Fig 3. The associated relationships for the apparent values of J, $F_1$, MCC and prevalence with prevalence are shown in Fig 4. The true values for each metric arising from the use of a gold reference standard are plotted in Figs 3 and 4 for comparison. Throughout, the classification being assessed had Recall = Specificity = 0.8.

The relationship between the apparent accuracy and prevalence for the four basic metrics of classification accuracy generated from the use of three imperfect reference data sets of differing accuracy associated with inclusion of correlated errors is shown in Fig 5. The associated relationships for the apparent values of J, $F_1$, MCC and prevalence with prevalence are shown in Fig 6. Again, the true values for each metric arising from the use of a gold reference standard are plotted in Figs 5 and 6 for comparison. Throughout, the classification being assessed had Recall = Specificity = 0.8.

Fig 7 shows the variation in true and apparent MCC with prevalence for the poor classification (Recall = Specificity = 0.5, and hence J = 0) assessed relative to an imperfect reference standard containing correlated errors (Recall = Specificity = 0.7, and hence J = 0.4).

Finally, to illustrate the effects of variations in prevalence and reference data quality on other metrics that are founded upon the core set of metrics discussed, the LR+ and LR-, which are based on Recall and Specificity, were also calculated. Fig 8 illustrates the variation in apparent LR+ and LR- values with prevalence obtained with the use of the reference standards of varying quality.

## 4. Discussion

For any simulated situation constructed, variation in prevalence and the use of an imperfect reference standard could make substantial changes to the confusion matrix from the truth. Consequently, the values of accuracy metrics and prevalence calculated from an apparent confusion matrix could deviate from the true situation.

Fig 2 summarises the key issues at two levels of prevalence: 0.1 and 0.3. In both cases, the classes were imbalanced, with positive cases rarer than negatives. Note that from the scenario used to drive the simulations the use of a gold standard reference would show Recall = Specificity = 0.8 (and J = 0.6). However, it is evident that the dissimilarities between Fig 2A and 2B show variation in confusion matrix values and metrics derived from them due to the difference in prevalence. Moreover, in both Fig 2A and 2B the use of an imperfect reference resulted in the apparent accuracy values deviating from the truth, the magnitude and

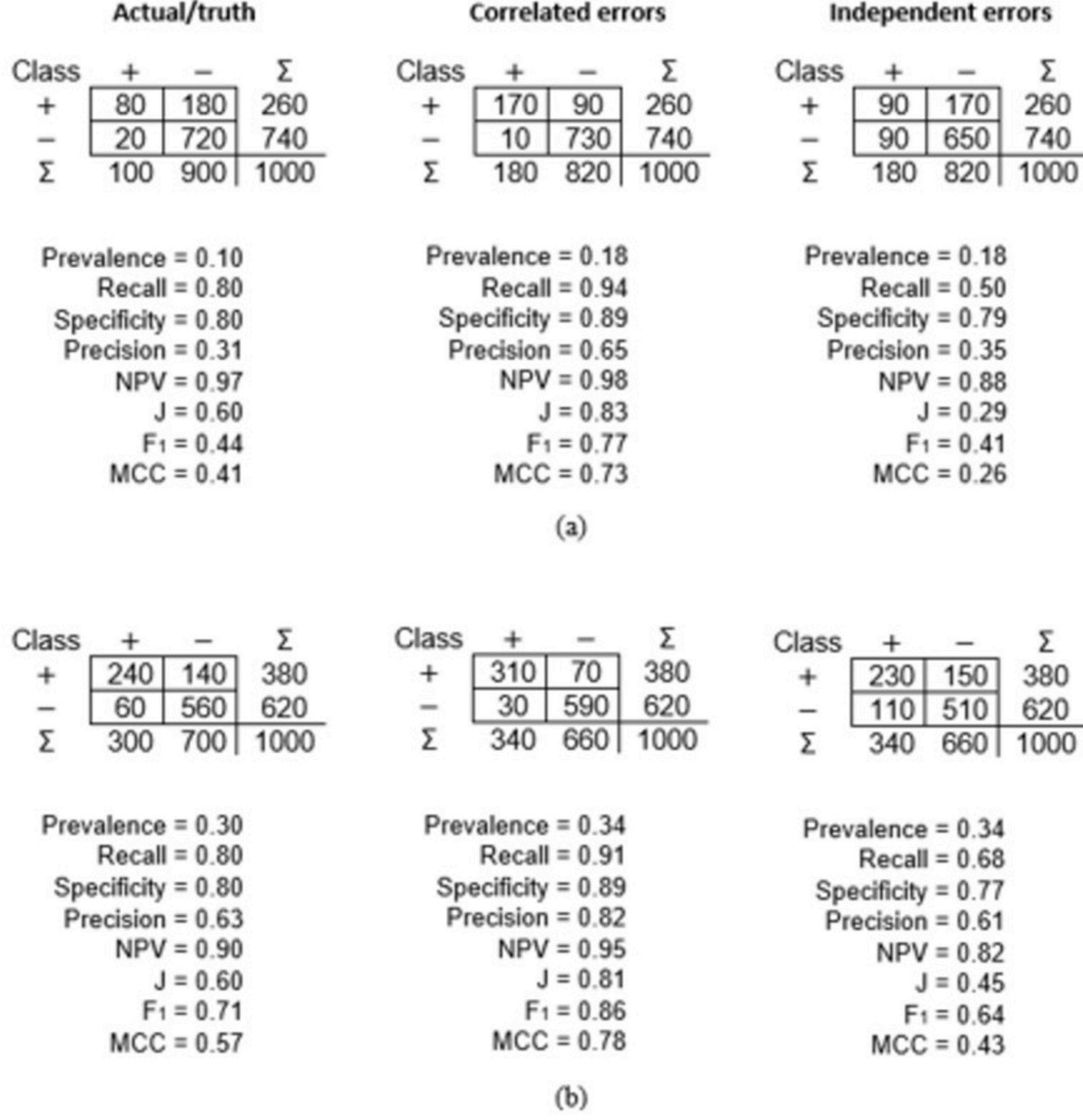

**Fig 2. Confusion matrices arising from the use of a gold standard reference, an imperfect reference with correlated errors (accuracy = 0.90) and an imperfect reference with independent errors (accuracy = 0.90).** (a) Prevalence = 0.1 and (b) Prevalence = 0.3.

direction of which differed between situations in which the errors were correlated or independent.

Critically, aside from the row marginal values (TP+FP and FN+TN) and total sample size (N), all of which were fixed, the value for every other element of the confusion matrix and the column marginal values could change with variation in prevalence and the use of an imperfect reference standard. It was, therefore, unsurprising that the values for the accuracy metrics calculated from the apparent confusion matrices differed from the truth. In the limited example provided in Fig 2 it is evident that the magnitude of mis-estimation varied greatly between the

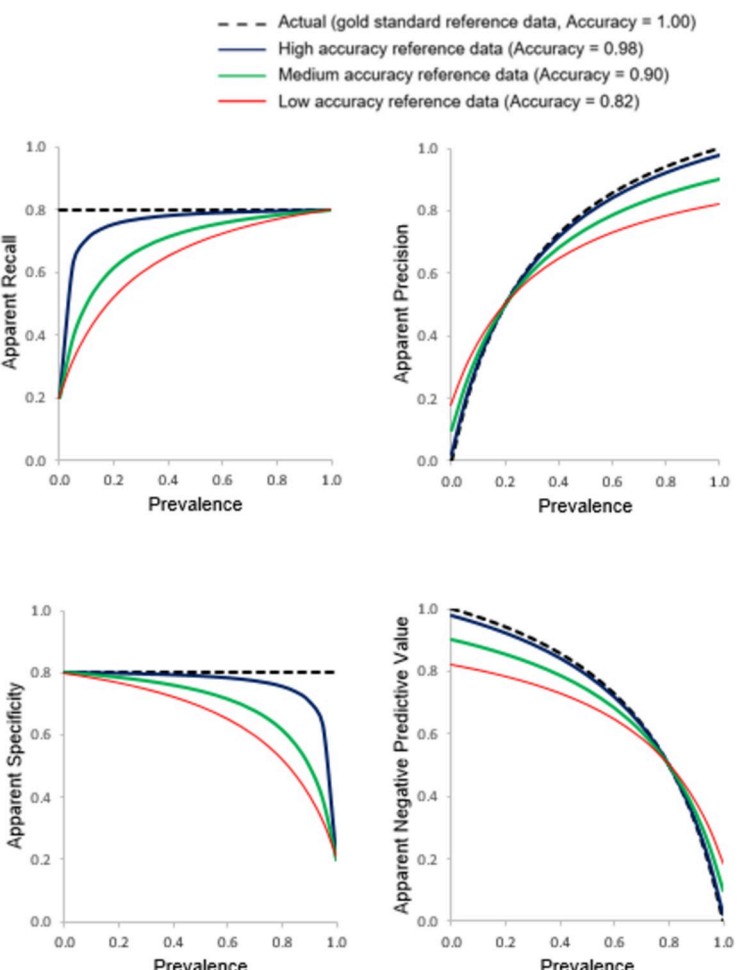

**Fig 3. Relationships between apparent Recall, Precision, Specificity and Negative Predictive Value with prevalence assessed using three imperfect reference standards of differing quality that contain errors independent of those in the classification.** The true relationship, obtained using a gold standard reference, is also shown for comparative purposes with a dashed black line.

various metrics calculated. For some accuracy metrics, the magnitude of mis-estimation was very small. For example, the Specificity when prevalence was 0.1 and the errors independent was calculated to be 0.79 while the true value was 0.80 (Fig 2A). Conversely, the value of some other metrics was greatly mis-estimated. For example, the Precision when the prevalence was 0.1 and the errors correlated was 0.65 while the true value was less than half of this value at 0.31 (Fig 2A).

Fig 2 provides a basis to illustrate some of the key impacts of reference data error and variation in prevalence on the confusion matrix and the metrics calculated from it. In the scenario in Fig 2A, the actual prevalence is 0.1. If, as shown in Fig 2A, an imperfect reference standard (Recall = Specificity = 0.9) was used the way the sample of cases was distributed in the matrix was changed from the true situation. Specifically, the values in all four matrix elements and the column marginal values could change; the row marginal values and total sample size were fixed. As Fig 2A shows, the use of the imperfect reference resulted in 90 (calculated from 100x0.9) of the 100 cases that truly were positive being labelled positive with the remaining 10 cases labelled as negative. Similarly, 810 (900x0.9) of the cases that truly were negative would

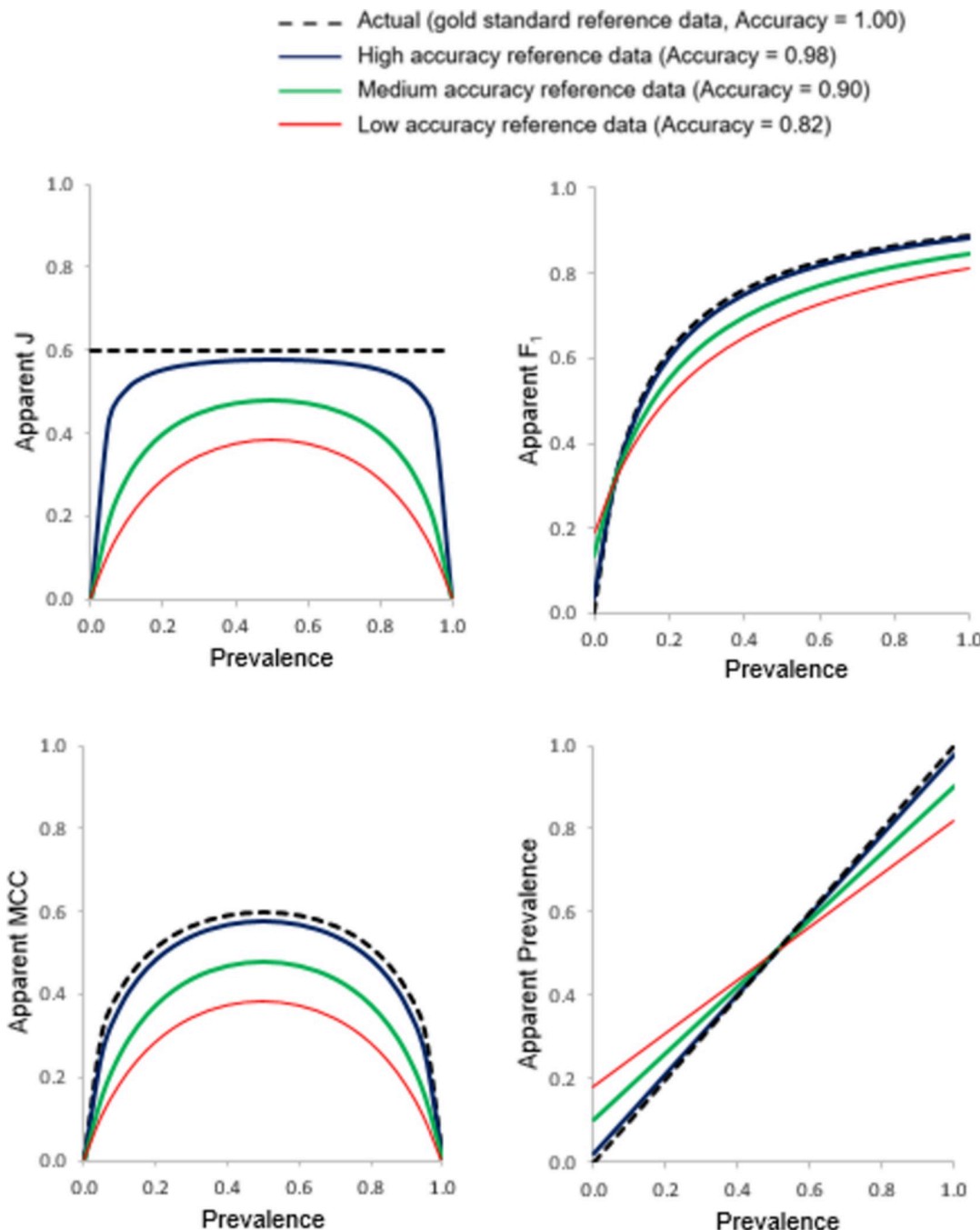

**Fig 4. Relationships between apparent J, $F_1$, MCC and prevalence with prevalence assessed using three imperfect reference standards of differing quality that contain errors independent of those in the classification.** The true relationship, obtained using a gold standard reference, is also shown for comparative purposes with a dashed black line.

be allocated to the negative class with the remaining 90 cases labelled positive. Hence, the column marginal values became 100−10+90 = 180 and 900+10−90 = 820 for the positive and negative class respectively. Consequently, an initial impact of the use of the imperfect reference was that the apparent prevalence rose from the true value to 0.18. The distribution of cases in the confusion matrix differed between the situations in which the errors were independent

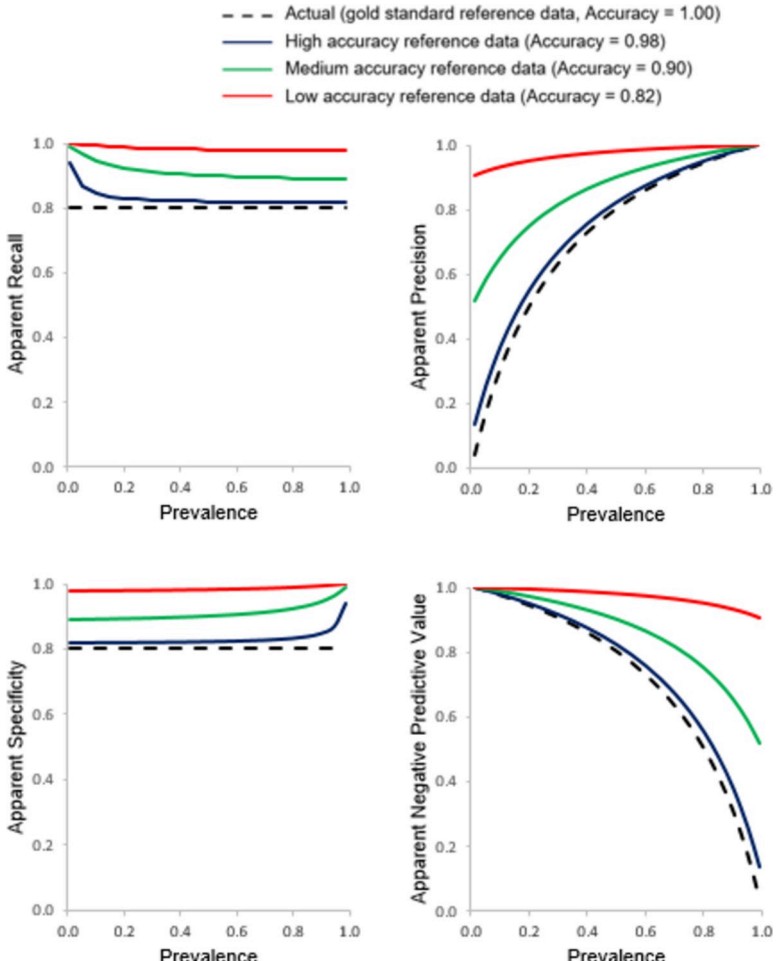

**Fig 5. Relationships between apparent Recall, Precision, Specificity and Negative Predictive Value with prevalence assessed using three imperfect reference standards of differing quality that contain errors correlated with those in the classification.** The true relationship, obtained using a gold standard reference, is also shown for comparative purposes with a dashed black line.

and correlated. Consequently, the accuracy metrics calculated from the confusion matrices associated with the use of the imperfect reference depended on the nature of the errors it contains.

The differences between Fig 2A and 2B illustrated a sensitivity of the apparent accuracy metrics to variation in prevalence. Note that claims that a key set of metrics, namely Recall, Specificity, J and MCC, are independent of prevalence holds when a gold standard reference was used but is untenable when an imperfect reference standard was used. The magnitude of the apparent accuracy assessed with all four of these metrics was underestimated when the errors in the reference standard and classification were independent of each other. Conversely, the magnitude of each of these four metrics was over-estimated when the errors in the reference standard and classification were correlated. The differences between the two apparent confusion matrices generated with the use of the imperfect reference standards (e.g. Fig 2A) arose because of the way in which cases are distributed within them.

The accuracy of the classification being assessed in Fig 2A can be summarised by Recall = Specificity = 0.8. With the errors in the reference standard being independent of those

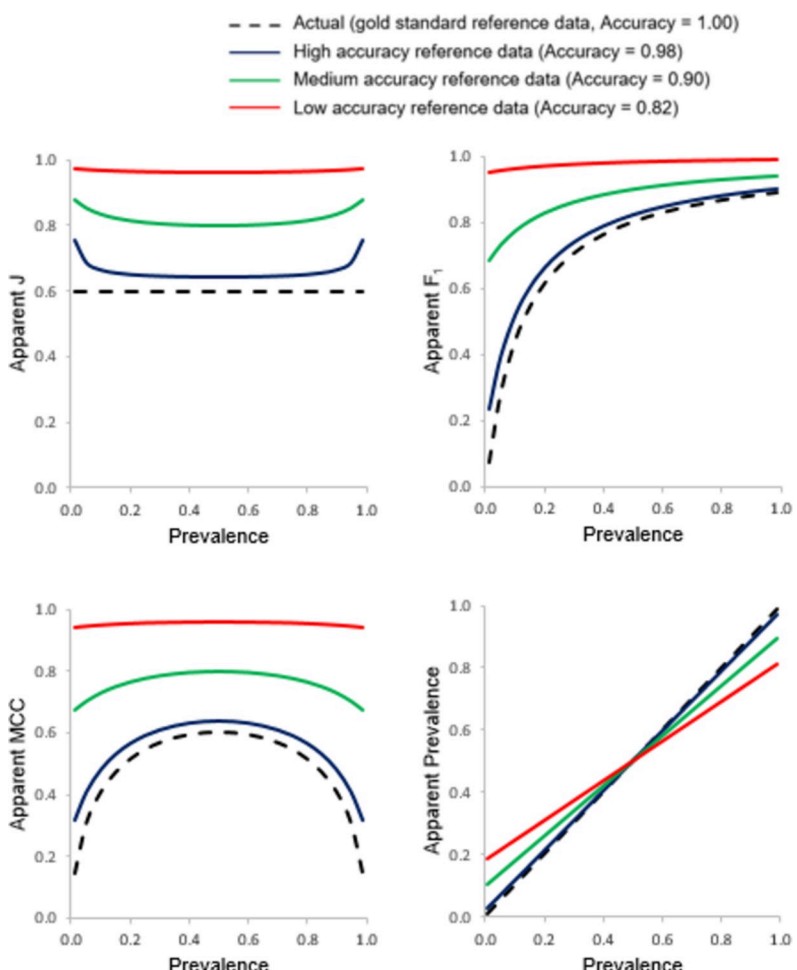

**Fig 6. Relationships between apparent J, F₁, MCC and prevalence with prevalence assessed using three imperfect reference standards of differing quality that contain errors correlated with those in the classification.** The true relationship, obtained using a gold standard reference, is also shown for comparative purposes with a dashed black line.

in the classification, the impacts of the use of this imperfect reference on the confusion matrix can be illustrated following the discussion in [30]. When the classifier was applied to the 90 truly positive cases, 72 (90x0.8) were labelled positive and the remaining 18 labelled as negative. For the 90 truly negative cases that were incorrectly labelled positive in the reference set 72 (90x0.8) remained labelled negative with the other 18 labelled positive. The net result of this situation was that $TP' = 72+18 = 90$ and $FN' = 18+72 = 90$. Similarly, of the 10 truly positive cases allocated to the negative class 8 (10x0.8) remained in the negative class with the remaining 2 cases labelled positive. Of the 810 cases truly negative cases 648 (810x0.8) remained as labelled negative with the other 162 labelled positive. Thus $FP' = 8+162 = 170$ and $TN' = 2+648 = 650$; the apparent confusion matrix values could also be calculated using Eqs 16–19. The cases were distributed differently within the confusion matrix when the errors in the reference standard were correlated with those in the classification.

The distribution of cases in the apparent confusion matrix generated when the imperfect reference standard contained errors correlated with those in the classification can be illustrated following the discussion in [16]. Maintaining a focus on the situation depicted in Fig 2A, the

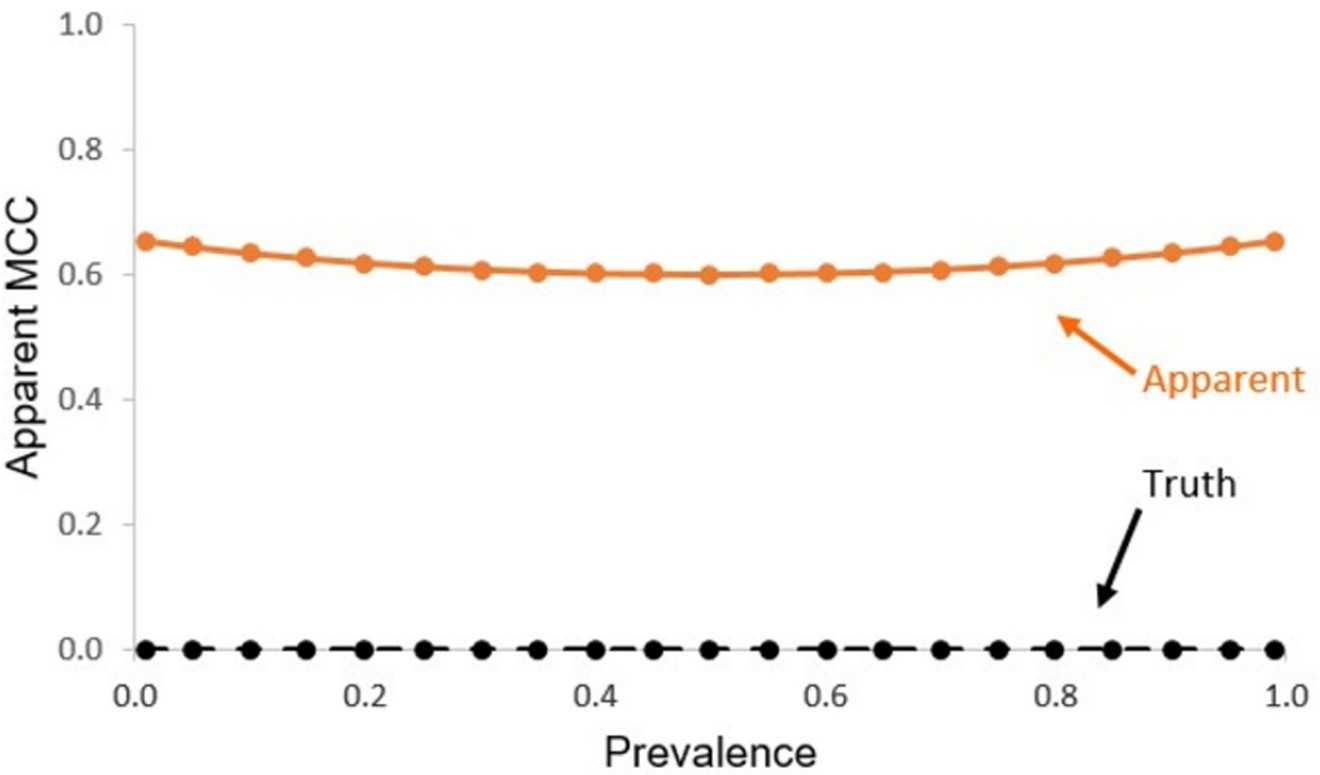

**Fig 7. Relationship of apparent MCC with prevalence for a poor classification (Recall = Specificity = 0.5, J = 0) assessed with an imperfect reference (Recall = Specificity = 0.7) containing correlated errors.** The true relationship, obtained using a gold standard reference, is also shown for comparative purposes with a dashed black line.

use of a reference containing correlated error is to alter the distribution of cases in the confusion matrix from the truth. While the column marginal values remained the same as in the situation when the errors were independent the distribution of cases within the confusion matrix could differ greatly. With correlated errors, the 90 truly negative cases that were labelled positive in both the reference and classification inflated TP' relative to the true value. Specifically, TP' = 80+90 = 170. Similarly, the 10 truly positive cases that were labelled negative in both the reference and classification inflated TN'; TN' = 720+10 = 730. Since the column marginal values are fixed at 180 and 820 for the positive and negative cases the values for FP' and FN' could be calculated. The net effect of this situation was that Recall and Specificity were over-estimated. Additionally, as captured in the differences between Fig 2A and 2B, the magnitude of mis-estimation varied with prevalence. Variation in the apparent values of the set of key accuracy metrics and prevalence over the full range of prevalence is explored below.

The magnitude of accuracy metrics beyond the set reported could also be expected to vary between scenarios. As one example, Accuracy (Eq 1) can be calculated from the confusion matrices shown in Fig 2. While the scenario adopted, in which the classification being assessed always had Recall = Specificity = 0.8 and hence Accuracy remains constant as prevalence varies, it is evident that the use of an imperfect reference resulted in Accuracy being mis-estimated. Specifically, while the simulation approach adopted ensures that the true value was always 0.80 the apparent values differ. With correlated errors, the Accuracy was over-estimated (0.90) while it was under-estimated if the errors were independent (0.74).

The nature of the difference between apparent and true values varied as a function of prevalence and the magnitude and nature of reference data error. If the errors in the reference

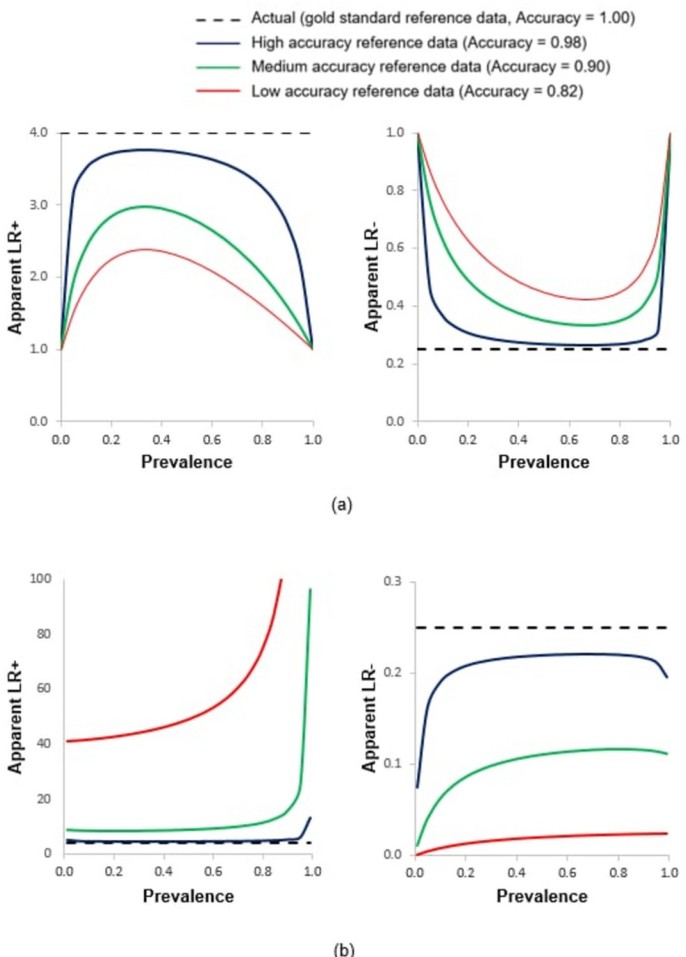

**Fig 8. Relationship between apparent LR+ and LR- values with prevalence assessed using three imperfect reference standards.** (a) Error in the reference is independent of that in the classification and (b) error in the reference is correlated with that in the classification. Note, in Fig 2B the Y axis for the positive LR was trimmed for visualisation purposes, the apparent value obtained for LR+ rises to 909.2. The true relationship, obtained using a gold standard reference, is also shown for comparative purposes with a dashed black line.

standard were independent it is evident that the apparent value for all four basic metrics of accuracy varied with prevalence (Fig 3). A key feature to note is that Recall and Specificity were no longer independent of prevalence and their magnitudes were under-estimated. Indeed, with the use of an imperfect reference standard containing independent errors, the apparent values for Recall and Specificity varied greatly with prevalence and were modulated by the magnitude of reference data error. Recall was substantially underestimated at low prevalence while Specificity was substantially under-estimated at high prevalence and the magnitude of mis-estimation was positively related to the size of the error in the reference data.

If attention was focused on the Precision and NPV, the apparent values of these metrics varied greatly with both prevalence and the magnitude of reference data error (Fig 3). For both Precision and NPV, the apparent values were under-estimated over part of the scale of prevalence and over-estimated for the remainder. The point of transition from under-to over-estimation occured at $(1\text{-}Specificity_R)/(2\text{-}Recall_R\text{-}Specificity_R)$ for Precision and $(1\text{-}Recall_R)/(2\text{-}Recall_R\text{-}Specificity_R)$ for the NPV [16]. The magnitude of mis-estimation was again positively related to the amount of error in the reference standard used.

The apparent values of J, $F_1$, MCC and prevalence all also varied with prevalence and the magnitude of reference data error (Fig 4). Since Recall and Specificity lost independence of prevalence due to the use of an imperfect reference standard so too did J. Indeed, the apparent value of J varied notably at the extremities of the scale of prevalence and for this metric the magnitude was always under-estimated. The estimates of apparent $F_1$ and prevalence also varied with prevalence and the magnitude of error-contained in the reference data set. Over most of the scale of prevalence, $F_1$ was underestimated. The apparent values of the MCC varied with prevalence, being particularly low at extreme values of prevalence. Again, the magnitude of mis-estimation was positively related to the degree of imperfection in the reference standard used. Of particular concern to this article is that very low, near zero, values for the MCC were obtained for a 'good' classification (Recall = Specificity = 0.8) if an imperfect reference standard was used and the data set was highly imbalanced. Such low values for the MCC could lead to the inappropriate decision to disregard the classification as being of insufficient accuracy when its actual accuracy could be adequate for the intended purpose. Finally, the apparent prevalence was linearly related to the actual prevalence and changed from over- to under-estimation as the actual prevalence increased. (Fig 4). At low prevalence, the apparent prevalence was substantially over-estimated (e.g. with 18% independent error, a prevalence of 0.01 was mis-estimated to be over 18 times too high). Conversely, at high prevalence the opposite trend was noted with prevalence under-estimated.

Different trends in the mis-estimation of accuracy metrics were observed when the reference standard contained correlated rather than independent errors. With the use of an imperfect reference standard containing correlated errors, all of the evaluated metrics of accuracy were over-estimated (Figs 5 and 6).

For the four basic metrics that are often calculated (Recall, Precision, Specificity and NPV), the effect of using an imperfect reference standard with correlated errors was to generate optimistically biased estimates across the entire scale of prevalence and with the magnitude of mis-estimation positively related to the degree of error in the reference standard used (Fig 5). As with the case of independent errors, Recall and Specificity varied with prevalence if an imperfect reference was used. The magnitude of the mis-estimation for the four accuracy metrics was most marked for Precision and NPV at extreme levels of prevalence. Precision was over-estimated at low prevalence and NPV over-estimated at high prevalence.

Given the changes to the confusion matrix associated with changes in prevalence and/or reference data error it was unsurprising that other metrics that in some way build on them were impacted. Fig 6 shows the apparent values for four key metrics often calculated: J, $F_1$, MCC and the prevalence. Since the magnitude of Recall and Specificity were no longer independent of prevalence, J too varied with prevalence, especially at the extreme values of the scale of prevalence. The two popular measure of $F_1$ and MCC also showed substantial dependency on prevalence. In all cases the magnitude of J, $F_1$ and MCC were over-estimated relative to the truth, notably at one or both extremities of the scale of prevalence. A key issue to note is that very high values of MCC, up to 0.96, were observed with the use of the least accurate reference data set. Finally, the prevalence itself, which may be the key property a study seeks to estimate, was also substantially mis-estimated. The trend for apparent prevalence was the same as that observed with the use of an imperfect reference containing independent error.

The over-estimation of apparent MCC arising through the use of an imperfect reference standard containing correlated error was also evident in the analyses based on the unquestionably poor classification (Recall = Specificity = 0.5, J = 0). For this classification, the use of a gold standard reference would result in MCC = 0. The apparent MCC values, however, were substantially over-estimated, with apparent values of up to 0.65 observed, with a relatively

small degree of variation over the range of prevalence (Fig 7). Critically, a relatively high apparent MCC value could be obtained from an unquestionably poor classification.

Variation in prevalence and impacts arising from the use of an imperfect reference standard would be expected with other popular metrics of accuracy. For example, popular approaches based on the receiver operating characteristics curve or the precision-recall curve are based on the set of basic accuracy metrics and hence would also be impacted by variations in prevalence and reference data imperfections. Similarly, the LRs which are based on Recall and Specificity would be expected to vary with prevalence even though they are often claimed to be unaffected by it. This latter issue is illustrated in Fig 8 as an example of impacts on metrics beyond the core set assessed here. It was evident that the LRs varied with prevalence and the magnitude of mis-estimation was positively related to the amount of error in the reference standard used. The LRs under-estimated the quality of the classification when the errors in the reference standard were independent. Conversely, the LRs over-estimated the quality of the classification when the imperfect reference standard contained correlated errors.

Of key relevance to this paper in relation to the use of the MCC was that a very low or very high apparent MCC could be observed for a 'good' classification depending on the level of prevalence and the quality and nature of the reference standard used. Thus, for example, a high MCC score could potentially arise from a modest or even poor classification. Alternatively, a low apparent MCC value may not reflect the actual status of a 'good' classification. The use of apparent MCC values may unjustifiably lead researcher to believe a classification to be of very different quality to the true situation. The comment that in relation to the four basic metrics (Eqs 3–6) the "MCC generates a high score only if all four of them are high" [15, page 13] may be true if implicit assumption of the use of a gold reference standard holds but sadly this may often not be the case. Naïve use of the apparent MCC such as in direct comparison against some popular threshold value or against values from another classification analysis with different properties (e.g. prevalence) may lead to inappropriate and incorrect interpretation of classifications and incorrect decision making. Just as other accuracy metrics which have been over-sold [13, 60], the MCC has limitations which can result in mis-leading interpretations of classification quality. Indeed some researchers deliberately stress that they do not endorse the use of the MCC [61].

The fundamental concerns with variations in prevalence and imperfections in the reference standard on accuracy estimation are well known and many have called for the issues to recognise and addressed. Rather than naively use apparent values researchers are instead encouraged to correct the assessment and estimate true values for accuracy metrics and prevalence [2, 16, 29, 30]. In some situations, such as when the reference standard contains independent errors but is of known quality, simple equations may be used to obtain the true values for accuracy metrics and prevalence [16, 30]. If the quality of the reference standard is not fully known it may also sometimes be possible to estimate them allowing the generation of truer values [62]. While correction for independent errors is easier than for correlated errors means to estimate truer values exist [16, 17]. In addition, for both independent and correlated errors, it is possible to effectively construct a reference standard by perhaps using the outputs of multiple classifications in a latent class analysis to estimate properties such as the Recall and Specificity of the classifications [20, 63, 64]. It is important that the calls to address the challenges associated with prevalence and use of imperfect reference data lead to change in the way classifications are routinely assessed and used. Researchers need to avoid bad habits such as the routine and unquestioning use of inappropriate metrics [60] especially if subject to mis-estimation due to commonly encountered challenges.

Finally, this article has focused on issues connected with classification accuracy assessment but the challenges associated with variations in prevalence and reference data error also impact

upon other aspects of a classification analysis. Prevalence and reference data error also impact on activities such as the training of supervised classifications and classifier development. The training of machine learning methods , for example, is impacted greatly by class imbalances [65] and hence the composition of the sample used in cross-validation should be carefully selected perhaps with regard to their abundance in the population under study [66]. Error in the reference standard can also degrade training data and ultimately classification performance and accuracy [67]. Further complications to accuracy assessment can also arise if other fundamental assumptions that underlie the analysis are unsatisfied. The conventional confusion matrix, for example, cannot be formed if the assumption that each case belongs fully to a single class in untenable. In such circumstances, a soft or fuzzy approach to accuracy assessment is required [68–70].

## 5. Conclusions

Classification analyses are widely used in a diverse array of disciplines. A fundamental issue in the use of a classification is its quality that is assessed typically via analysis of a confusion matrix that cross tabulates the labels predicted by the classification against those obtained from a reference standard for a sample of cases. The calculation of accuracy metrics and associated variables such as prevalence from the confusion matrix is, however, fraught with challenges.

The literature promotes a wide range of accuracy metrics and a common concern is that each typically only provides partial information on classification quality. Additionally, the magnitude of some metrics may vary with prevalence that is not a property of the classification but of the population under study. It is common to see researchers encouraged to use one or more metrics that are claimed to be prevalent independent (e.g. Recall, Specificity and J). Such metrics are widely used but do not fully capture the entire quality of a classification. Recent literature has encouraged the use of the MCC. The latter has been claimed to be a more truthful accuracy metric than other popular methods. Additionally, if prevalence is a concern with the use of the MCC it has been suggested that J could be used to summarise key aspects of classification quality. As with other accuracy metrics, however, important challenges arise in real world applications. An uncomfortable truth is that the reference standard used to assess accuracy is often imperfect and hence a fundamental assumption in classification accuracy assessment that is often made implicitly is unsatisfied. This is well known but rarely checked or addressed.

Here, the effect of variations in prevalence and reference standard error are shown to substantially impact on the assessment of classification accuracy and calculation of properties such as class abundance. Accuracy metrics such as Recall, Specificity and J lost their independence of prevalence when an imperfect reference standard was used in the accuracy assessment. Indeed, all of the accuracy metrics included in this article (Recall, Precision, Specificity, NPV, J, $F_1$, LR+, LR- and MCC) were sensitive to variations in prevalence and use of an imperfect reference standard. Critically, the estimated values of accuracy metrics deviated substantially from the true values that would be obtained by the use of a gold standard reference.

The magnitude and direction of mis-estimation was notably a function of prevalence and the size and nature of the imperfections in the reference standard. For example, the four basic metrics of accuracy (Recall, Precision, Specificity and NPV) were all under-estimated when the reference standard contained errors that were independent of the classification. However, when the errors in the reference standard were correlated, that is tending to err on the same cases the classifier mis-labelled, the opposite trend was observed with the values of all of the accuracy metrics over-estimated.

Of particular importance, however, is that the MCC displayed undesirable properties. It was possible for the apparent value of the MCC to be substantially under- or over-estimated as a result of variations in prevalence and/or use of an imperfect reference. Critically, a high value for the MCC could be obtained from a poor classification, notably if the reference standard used was inaccurate with errors correlated with those in the classification under evaluation. This observation runs contrary to arguments promoting the MCC that contend that a high value is only possible when the classification has performed well on both classes. Such arguments are founded on unrealistic conditions (e.g. on the use of a gold standard reference). In reality, the magnitude of the MCC is influenced greatly by class imbalance and reference data imperfections.

Real world challenges to accuracy assessment need to be addressed. Many of the fundamental issues are well known but rarely acted on. Researchers should recognise the problems and take action to address them such as by estimating true values. This may require a culture change as methods and practices often seem to be fixed firmly and research communities resistant to change. However, if classifications are to be evaluated and used appropriately the routine use of inappropriate metrics and mis-interpretation of apparent values must stop. Readers of published studies based on a classification analysis should also interpret results and associated interpretations with care, especially if no explicit account is made for the effects of prevalence and reference data error. This is not to question the integrity of the authors of a study but simply to recognise the possible impacts arising from a failure to satisfy an underlying assumption in an accuracy assessment. Fortunately, a range of methods exist to address the problems that arise from class imbalance and use of an imperfect reference in order to allow an enhanced evaluation of classifications [20, 63, 71].

## Supporting information

**S1 Table. Confusion matrix elements for scenarios with independent errors.**
(DOCX)

**S2 Table. Confusion matrix elements for scenarios with correlated errors.**
(DOCX)

**S3 Table. Confusion matrix elements for the scenario focused on a poor classification.**
(DOCX)

## Acknowledgments

I am grateful to the three referees for their helpful comments that helped enhance this article. The work reported here benefit from a period of research leave provided by the School of Geography, University of Nottingham.

## Author Contributions

**Conceptualization:** Giles M. Foody.

**Formal analysis:** Giles M. Foody.

**Investigation:** Giles M. Foody.

**Methodology:** Giles M. Foody.

**Project administration:** Giles M. Foody.

**Writing – original draft:** Giles M. Foody.

**Writing – review & editing:** Giles M. Foody.

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
