## [Decision Letter · Decision Letter 0]

23 May 2023

PONE-D-23-13169Challenges in the real world use of classification accuracy metrics: from recall and precision to the Matthews correlation coefficientPLOS ONE

Dear Dr. Authors,

Thank you for submitting your manuscript to PLOS ONE. After careful consideration, we feel that it has merit but does not fully meet PLOS ONE’s publication criteria as it currently stands. Therefore, we invite you to submit a revised version of the manuscript that addresses the points raised during the review process.

ACADEMIC EDITOR: Major RevisionPlease ensure that your decision is justified on PLOS ONE’s publication criteria and not, for example, on novelty or perceived impact.

We look forward to receiving your revised manuscript.

Kind regards,

Shigao Huang

Academic Editor

PLOS ONE

Journal Requirements:

Reviewers' comments:

Reviewer's Responses to Questions

**Comments to the Author**

1. Is the manuscript technically sound, and do the data support the conclusions?

Reviewer #1: Partly

Reviewer #2: Partly

2. Has the statistical analysis been performed appropriately and rigorously? 

Reviewer #1: No

Reviewer #2: No

3. Have the authors made all data underlying the findings in their manuscript fully available?

Reviewer #1: No

Reviewer #2: No

4. Is the manuscript presented in an intelligible fashion and written in standard English?

Reviewer #1: Yes

Reviewer #2: Yes

5. Review Comments to the Author

Reviewer #1: Reviewer’s Concern # 1: First of all, include the introductory sentence about the problem statement in abstract.

Reviewer’s Concern # 2: In the second part, the author should give an overall description of the entire technique first, then give the point-by-point introduction to these used techniques.

Reviewer’s Concern # 3: The author should define all performance measures first, then compare the results of different schemes.

Reviewer’s Concern # 4: What is the research significance of this article?

Reviewer’s Concern # 5: There is little literature in the past three years, and the latest research should be cited.

Reviewer’s Concern # 6: Correct the references and follow the reference format of PLOS ONE.

Reviewer’s Concern # 7: The authors should include a more extensive discussion of the novelty of their system in Introduction Part.

Reviewer’s Concern # 8: The English language needs to be seriously improved. Some sentences are incomplete, others have different meaning than the authors intended to. Also, the authors should try to shorten some of the sentences. Many of them are too long and are not easy to follow and understand.

Reviewer’s Concern # 9: The structure of the paper is, in my opinion, too confusing, too complex. The introduction should be shorter and much and more to the point. It fails to do the one of the most important roles of the introduction, it fails to present sufficient information, so that important readers and evaluators do not need to read the rest of the text, being assured in the contribution made, and validity of the text to follow. It should establish the paper’s motivation, methods, contribution, results, and position in scientific literature. For each of the sections, similar comments could be made.

Reviewer’s Concern # 10: Include Discussion section, which is missing.

Reviewer’s Concern # 11: The Conclusion is very lengthy. Briefly elaborate it in a paragraph.

Reviewer #2: While the topic addressed in this paper is certainly important, I feel the treatment of the details was somewhat underwhelming. Specifically, I would urge the author to add the following to the manuscript:

1.Reading through the paper, there were minimal and negligible insights into why the different parameters behave the way they do. For instance, sensitivity and specificity are theoretically defined within two subsets of the population - positives and negatives based on the reference standards. In that case, these parameters should be theoretically independent of prevalence. Yet, in practice we do know (as the author also notes) that sensitivity and specificity are affected or influenced by prevalence. A more detailed description of why such behavior is seen is what I was really looking forward to.

2. In clinical epidemiology, the concept of positive and negative likelihood ratios is fairly well established. The LRs provide a clinically meaningful interpretation that is minimally influenced by prevalence. I did not find any discussion or reference to the LRs in the paper. I suggest that the author add LRs to the same data used in the paper and demonstrate the relative robustness of these parameters in classification tasks.

3. I feel the paper does not end with specific recommendations for authors. That the use of these parameters is fraught with dangers is well known. What a researcher actually needs to do when confronted with the situation is not clear from the paper. The value of the paper will be greatly improved if such recommendations - based on real or simulated data - can be categorically spelt out.

6. PLOS authors have the option to publish the peer review history of their article (what does this mean?). If published, this will include your full peer review and any attached files.

Reviewer #1: No

Reviewer #2: No

---

## [Author Response · Author response to Decision Letter 0]

9 Jun 2023

Please see the 'Response to Reviewers' document uploaded with the manuscript files.

---

## [Decision Letter · Decision Letter 1]

23 Aug 2023

PONE-D-23-13169R1Challenges in the real world use of classification accuracy metrics: from recall and precision to the Matthews correlation coefficientPLOS ONE

Dear Dr. Foody,

Thank you for submitting your manuscript to PLOS ONE. After careful consideration, we feel that it has merit but does not fully meet PLOS ONE’s publication criteria as it currently stands. Therefore, we invite you to submit a revised version of the manuscript that addresses the points raised during the review process.

We look forward to receiving your revised manuscript.

Kind regards,

Shigao Huang

Academic Editor

PLOS ONE

Journal Requirements:

Reviewers' comments:

Reviewer's Responses to Questions

**Comments to the Author**

1. If the authors have adequately addressed your comments raised in a previous round of review and you feel that this manuscript is now acceptable for publication, you may indicate that here to bypass the “Comments to the Author” section, enter your conflict of interest statement in the “Confidential to Editor” section, and submit your "Accept" recommendation.

Reviewer #1: All comments have been addressed

Reviewer #3: (No Response)

2. Is the manuscript technically sound, and do the data support the conclusions?

Reviewer #1: Yes

Reviewer #3: Yes

3. Has the statistical analysis been performed appropriately and rigorously? 

Reviewer #1: Yes

Reviewer #3: N/A

4. Have the authors made all data underlying the findings in their manuscript fully available?

Reviewer #1: No

Reviewer #3: Yes

5. Is the manuscript presented in an intelligible fashion and written in standard English?

Reviewer #1: Yes

Reviewer #3: Yes

6. Review Comments to the Author

Reviewer #1: Some minor edits required in spellings and grammar mistakes and also correct the references style according to PLOS ONE journal. You may cite latest articles in your manuscript, e.g.

1. https://doi.org/10.1371/journal.pone.0240015

2. https://doi.org/10.3390/s22072724

3. https://doi.org/10.1109/ACCESS.2021.3084905

Reviewer #3: While I was not a reviewer for the first version of the manuscript, I was invited to review Revision 1. I believe the author has adequately addressed all comments from the original reviewers. My recommendation of "Minor Revision" arises from a sentence in Rev1. Lines 763-765 state that this article has shown that a high MCC value does not necessarily imply good classification performance in both classes of a binary classification problem. However, lines 689-691 explicitly mention that this is to be expected in the case of a gold standard reference. Hence, the conclusions should be changed to indicate that MCC will have poor performance with inaccurate datasets and grossly imbalanced classes.

Line 690 should be changed from "all four of them area high” to "all four of them are high"

Finally, this article misses specific recommendations to address the shortcomings of classification metrics when the reference data has errors. The article mentions many times that imperfect reference data will adversely affect the classification metrics, but in that case even the confusion matrices will be unreliable. The author should share some insights on how to address this situation, since the article is aimed at the practitioner.

7. PLOS authors have the option to publish the peer review history of their article (what does this mean?). If published, this will include your full peer review and any attached files.

Reviewer #1: No

Reviewer #3: No

---

## [Author Response · Author response to Decision Letter 1]

27 Aug 2023

A file with the response to reviewer comments has been uploaded. It is copied here:

Response to reviewer’s comments

Thank you for the comments on the revised article. These comments have informed the further revision of the article and below a brief response is provided to each point. For completeness and clarity, the full set of comments received is copied below and a brief response made on an essentially point-by-point basis. The responses are provided in red italics to help separate them from the comments. 

Comments to the Author

6. Review Comments to the Author

Reviewer #1: Some minor edits required in spellings and grammar mistakes and also correct the references style according to PLOS ONE journal. You may cite latest articles in your manuscript, e.g.

1. https://doi.org/10.1371/journal.pone.0240015

2. https://doi.org/10.3390/s22072724

3. https://doi.org/10.1109/ACCESS.2021.3084905

Thank you for the comments. Some minor editing has been undertaken. The suggested references are interesting and use some of the metrics discussed in the article but their inclusion would not add constructively; the suggested articles have not been cited.

Reviewer #3: While I was not a reviewer for the first version of the manuscript, I was invited to review Revision 1. I believe the author has adequately addressed all comments from the original reviewers. My recommendation of "Minor Revision" arises from a sentence in Rev1. Lines 763-765 state that this article has shown that a high MCC value does not necessarily imply good classification performance in both classes of a binary classification problem. However, lines 689-691 explicitly mention that this is to be expected in the case of a gold standard reference. Hence, the conclusions should be changed to indicate that MCC will have poor performance with inaccurate datasets and grossly imbalanced classes.

Thank you for the comment. I agree that highlighting the use of an imperfect reference is critical here and hence the text in the conclusions has been revised. The relevant paragraph in the (clean) version of the manuscript on lines 760-768 now reads:

“Of particular importance, however, is that the MCC displayed undesirable properties. It was possible for the apparent value of the MCC to be substantially under- or over-estimated as a result of variations in prevalence and/or use of an imperfect reference. Critically, a high value for the MCC could be obtained from a poor classification, notably if the reference standard used was inaccurate with errors correlated with those in the classification under evaluation. This observation runs contrary to arguments promoting the MCC that contend that a high value is only possible when the classification has performed well on both classes. Such arguments are founded on unrealistic conditions (e.g. on the use of a gold standard reference). In reality, the magnitude of the MCC is influenced greatly by class imbalance and reference data imperfections.”

The text highlighted in bold above aims to address the point flagged by the referee.

Line 690 should be changed from "all four of them area high” to "all four of them are high"

The change has been made.

Finally, this article misses specific recommendations to address the shortcomings of classification metrics when the reference data has errors. The article mentions many times that imperfect reference data will adversely affect the classification metrics, but in that case even the confusion matrices will be unreliable. The author should share some insights on how to address this situation, since the article is aimed at the practitioner.

This is a good point but discussion on the methods to address the problems would be a whole new paper it itself. However, I agree with the referee that it would be useful to point to means to address the problems. The very final sentence of the concluding section has been changed and points the interested reader to 3 articles that discuss several approaches that may be used – specifically to references 20, 63 and 71 (the latter is a new reference added as a result of the referee’s comments).

I am grateful to the referees and editorial team for their comments on the article.

---

## [Decision Letter · Decision Letter 2]

11 Sep 2023

Challenges in the real world use of classification accuracy metrics: from recall and precision to the Matthews correlation coefficient

PONE-D-23-13169R2

Dear Dr. Foody,

We’re pleased to inform you that your manuscript has been judged scientifically suitable for publication and will be formally accepted for publication once it meets all outstanding technical requirements.

Kind regards,

Shigao Huang

Academic Editor

PLOS ONE

Additional Editor Comments (optional):

Reviewers' comments:

Reviewer's Responses to Questions

**Comments to the Author**

1. If the authors have adequately addressed your comments raised in a previous round of review and you feel that this manuscript is now acceptable for publication, you may indicate that here to bypass the “Comments to the Author” section, enter your conflict of interest statement in the “Confidential to Editor” section, and submit your "Accept" recommendation.

Reviewer #3: All comments have been addressed

2. Is the manuscript technically sound, and do the data support the conclusions?

Reviewer #3: Yes

3. Has the statistical analysis been performed appropriately and rigorously? 

Reviewer #3: N/A

4. Have the authors made all data underlying the findings in their manuscript fully available?

Reviewer #3: No

5. Is the manuscript presented in an intelligible fashion and written in standard English?

Reviewer #3: Yes

6. Review Comments to the Author

Reviewer #3: (No Response)

7. PLOS authors have the option to publish the peer review history of their article (what does this mean?). If published, this will include your full peer review and any attached files.

Reviewer #3: No

---

## [Editor Report · Acceptance letter]

25 Sep 2023

PONE-D-23-13169R2 

Challenges in the real world use of classification accuracy metrics: from recall and precision to the Matthews correlation coefficient 

Dear Dr. Foody:

I'm pleased to inform you that your manuscript has been deemed suitable for publication in PLOS ONE. Congratulations! Your manuscript is now with our production department. 

Kind regards, 

on behalf of

Dr. Shigao Huang 

Academic Editor

PLOS ONE